# Beta-Adrenergic Activation of the Inward Rectifier K^+^ Current Is Mediated by the CaMKII Pathway in Canine Ventricular Cardiomyocytes

**DOI:** 10.3390/ijms252111609

**Published:** 2024-10-29

**Authors:** Zsigmond Máté Kovács, Balázs Horváth, Csaba Dienes, József Óvári, Dénes Kiss, Tamás Hézső, Norbert Szentandrássy, János Magyar, Tamás Bányász, Péter Pál Nánási

**Affiliations:** 1Department of Physiology, Faculty of Medicine, University of Debrecen, 4032 Debrecen, Hungary; kovacs.zsigmond@med.unideb.hu (Z.M.K.); dienes.csaba@med.unideb.hu (C.D.); ovari.jozsef@med.unideb.hu (J.Ó.); kiss.denes@med.unideb.hu (D.K.); hezso.tamas@med.unideb.hu (T.H.); szentandrassy.norbert@med.unideb.hu (N.S.); magyar.janos@med.unideb.hu (J.M.); banyasz.tamas@med.unideb.hu (T.B.); nanasi.peter@med.unideb.hu (P.P.N.); 2Department of Basic Medical Sciences, Faculty of Dentistry, University of Debrecen, 4032 Debrecen, Hungary; 3Division of Sport Physiology, Department of Physiology, Faculty of Medicine, University of Debrecen, 4032 Debrecen, Hungary; 4Department of Dental Physiology and Pharmacology, Faculty of Dentistry, University of Debrecen, 4032 Debrecen, Hungary

**Keywords:** inward rectifier K^+^ current, rapid delayed rectifier K^+^ current, sympathetic activation, protein kinase A, calcium/calmodulin-dependent protein kinase II, action potential voltage clamp, canine cardiomyocytes

## Abstract

Several ion currents in the mammalian ventricular myocardium are substantially regulated by the sympathetic nervous system via β-adrenergic receptor activation, including the slow delayed rectifier K^+^ current and the L-type calcium current. This study investigated the downstream mechanisms of β-adrenergic receptor stimulation by isoproterenol (ISO) on the inward rectifier (I_K1_) and the rapid delayed rectifier (I_Kr_) K^+^ currents using action potential voltage clamp (APVC) and conventional voltage clamp techniques in isolated canine left ventricular cardiomyocytes. I_K1_ and I_Kr_ were dissected by 50 µM BaCl_2_ and 1 µM E-4031, respectively. Acute application of 10 nM ISO significantly increased I_K1_ under the plateau phase of the action potential (0–+20 mV) using APVC, and similar results were obtained with conventional voltage clamp. However, β-adrenergic receptor stimulation did not affect the peak current density flowing during terminal repolarization or the overall I_K1_ integral. The ISO-induced enhancement of I_K1_ was blocked by the calcium/calmodulin kinase II (CaMKII) inhibitor KN-93 (1 µM) but not by the protein kinase A inhibitor H-89 (3 µM). Neither KN-93 nor H-89 affected the I_K1_ density under baseline conditions (in the absence of ISO). In contrast, parameters of the I_Kr_ current were not affected by β-adrenergic receptor stimulation with ISO. These findings suggest that sympathetic activation enhances I_K1_ in canine left ventricular cells through the CaMKII pathway, while I_Kr_ remains unaffected under the experimental conditions used.

## 1. Introduction

Sympathetic activation of cardiac tissues is the primary mechanism by which the heart adapts to “fight-or-flight” situations. This leads to characteristic changes in contractility, calcium handling, and the morphology of the action potential, the latter due to modifications in underlying ion currents. Many cardiac ion currents, particularly the L-type calcium current (I_Ca,L_) and the slow delayed rectifier potassium current (I_Ks_), are robustly enhanced by sympathetic stimulation through ß-adrenergic receptors. However, the effects of sympathetic stimulation on the ion currents that are responsible for the terminal repolarization of the action potential, such as the inward rectifier K^+^ current (I_K1_) and the rapid delayed rectifier K^+^ current (I_Kr_), remain poorly understood.

Isoproterenol (ISO, 1 µM), the well-known ß-adrenergic receptor agonist, suppressed I_K1_ in guinea pig ventricular myocytes [1] and canine Purkinje cells [2], while 10 nM ISO augmented I_K1_ in murine myocytes [3]. Not only are the effects inconsistent, but the underlying mechanisms are also controversial. According to the classic theory, most cardiac effects of catecholamines are mediated via the ß_1_ receptor—cAMP—protein kinase A (PKA) pathway. However, the effect of PKA on I_K1_ remains disputed, as studies have shown that I_K1_ and Kir2.1 (KCNJ2) currents can either increase [4,5] or decrease [6,7] following PKA activation. Additionally, protein kinase C (PKC) activation generally appears to suppress I_K1_ [4,8] and its inhibition results in increased I_K1_ [9]. However, a different study reported that PKC activation increased I_K1_ [10].

I_Kr_ was increased in canine myocytes by ISO and PKA activators, an effect which was abolished in the presence of PKA inhibitors [11]. Similarly, ISO and the direct adenylyl cyclase activator forskolin increased I_Kr_ in guinea pig myocytes, effects which were sensitive to both PKA and PKC inhibition, as well as to a reduction in the intracellular calcium concentration ([Ca^2+^]_i_) caused by BAPTA-AM or nifedipine [12]. In contrast, both ß_1_ receptor activation [13] and phosphodiesterase inhibition [14] decreased I_Kr_ and K_v_11.1 (KCNH2; hERG) currents in a PKA-dependent mechanism in guinea pig myocytes.

Sympathetic activation raises [Ca^2+^]_i_, and Ca^2+^ ions may mediate certain effects of sympathetic activation in cardiac tissues through several mechanisms: (1) directly by themselves, (2) by forming complexes with Ca^2+^ binding proteins like calmodulin, and/or (3) via other Ca^2+^-sensitive regulatory pathways, most notably the calcium/calmodulin-dependent protein kinase II (CaMKII). Previous studies have demonstrated that intracellular Ca^2+^ exerts a dynamic inhibitory effect on I_K1_ at positive voltages [15,16]. Therefore, an increase in [Ca^2+^]_i_ during sympathetic activation is expected to enhance the rectification of I_K1_. Additionally, PKC might mediate indirect Ca^2+^-dependent reduction of I_K1_, because the activation of “conventional” Ca^2+^-dependent PKC isoforms (e.g., PKCβ) has been reported to decrease the inward component of I_K1_ in Xenopus oocytes, in rat and mouse models [8,17], and in human atrial cells [8]. However, it remains unclear whether PKC affects the physiologically important outward component of the I_K1_ current.

Another indirect Ca^2+^-dependent pathway, CaMKII, also influences I_K1_. In murine myocytes, I_K1_ was reduced after CaMKII overexpression and increased following chronic CaMKII inhibition [18,19]. In rabbits, however, the overexpression of CaMKII led to an increase in I_K1_ [19]. The same study also found that acute activation of CaMKII increased the I_K1_ density in both rabbits and rats [19]. Similarly, in canine ventricular cells, I_K1_ was elevated following a rise in [Ca^2+^]_i_, but this effect was blocked by KN-93, a CaMKII inhibitor [20].

These findings suggest that the mechanisms by which sympathetic activation modulates I_K1_ and I_Kr_ in cardiac cells are still debated and appear to strongly depend on the specific experimental conditions and species studied. This complexity is further compounded by the known interaction between the PKA and CaMKII pathways [21]. Therefore, this study aims to disentangle the contributions of the PKA- and CaMKII-mediated mechanisms in the β-adrenergic regulation of I_K1_ and I_Kr_ in canine ventricular myocytes, with an experimental setting that closely mimics physiological conditions. This was achieved using the action potential voltage clamp (APVC) technique, keeping the intracellular Ca^2+^ homeostasis intact (without buffering [Ca^2+^]_i_) and applying a non-saturating concentration of the β-adrenergic agonist (10 nM ISO). Canine myocytes were selected for this study, because their electrophysiological properties closely resemble those of human ventricular cells [22,23,24,25].

## 2. Results

### 2.1. The Effect of β-Adrenergic Receptor Activation on I_K1_

Parameters of I_K1_ were compared in five separate groups of myocytes under APVC conditions: (1) control (no treatment), (2) acute effect of 10 nM ISO, (3) acute effect of ISO in the presence of KN-93, (4) acute effect of ISO in the presence of H-89, and (5) acute effect of ISO in the presence of KN-93 + H-89. Figure 1 shows representative I_K1_ current traces obtained under these conditions. 10 nM ISO was used to achieve a biologically relevant moderate β-receptor activation, without receptor saturation. 

β-adrenergic receptor stimulation with ISO increased the mid-plateau density (current density measured at 50% of APD_90_) of I_K1_ significantly (0.067 ± 0.019 A/F in control vs. 0.159 ± 0.029 A/F in ISO, *n* = 7 in both groups; Figure 2A). This effect of ISO was prevented by pretreatment with 1 µM KN-93 (CaMKII inhibitor; 0.073 ± 0.014 A/F, *n* = 7) but not by 3 µM H-89 (PKA inhibitor; 0.136 ± 0.024 A/F, *n* = 10), as also shown in Figure 2A. When the cells were pretreated with the combination of KN-93 and H-89, the mid-plateau density of I_K1_ (0.065 ± 0.013 A/F, *n* = 9) was similar to the value obtained either under control conditions or after KN-93 pretreatment alone. Interestingly, no significant effect of ISO was detected when either the peak value of I_K1_ (1.849 ± 0.120 A/F in control vs. 1.967 ± 0.159 A/F in ISO), measured during terminal repolarization, or the total current integral of I_K1_ (66.7 ± 4.4 mC/F in control vs. 77.7 ± 8.1 mC/F in ISO) was analyzed (Figure 1 and Figure 2A–C).

These results indicate that the β-adrenergic-pathway-mediated augmentation of I_K1_ appears during the plateau of the AP; therefore, the density of I_K1_ was analyzed at two fixed membrane potential levels of +20 mV and 0 mV. The former is close to the mid-plateau potential, while the latter overlaps with a more accelerated level of repolarization (Figure 3A). As shown in the panels in Figure 3B,C, the pattern is identical to the one seen in Figure 2, i.e., the I_K1_ density was significantly larger in ISO (0.148 ± 0.034 A/F at +20 mV, 0.202 ± 0.041 A/F at 0 mV) compared to control conditions (0.042 ± 0.011 A/F and 0.090 ± 0.018 A/F, respectively), while this effect of ISO was prevented by KN-93 (0.052 ± 0.012 A/F and 0.095 ± 0.028 A/F, respectively) but not by H-89 (0.127 ± 0.025 A/F and 0.159 ± 0.022 A/F, respectively).

Similar results were obtained when the effects of ISO alone and in the presence of KN-93 or H-89 on I_K1_ density were studied under conventional voltage clamp conditions at test potentials ranging from −80 mV to +20 mV. Compared to the control cells, the I_K1_ density was significantly larger in ISO in the range of −30 mV to +20 mV, an effect which was abolished by CaMKII blockade with KN-93 but not by PKA inhibition with H-89 (Figure 4A–C). At 0 mV, the I_K1_ current densities were 0.266 ± 0.042 A/F in control conditions (n = 11), 0.525 ± 0.073 A/F in ISO-pretreated cells (n = 9), 0.211 ± 0.067 A/F in KN-93 + ISO (n = 10), and 0.431 ± 0.103 A/F in H-89 + ISO (n = 9), as illustrated in Figure 4C. No significant differences between the cell groups were observed at membrane potentials below −30 mV.

### 2.2. Effects of PKA and CaMKII Inhibition on I_K1_ Without β-Adrenergic Receptor Stimulation

The effects of KN-93 and H-89 on I_K1_ were also studied under baseline conditions, without β-adrenergic receptor activation. No differences were observed in the mid-plateau density, peak current density, or current integral between the untreated group and those exposed to KN-93 or H-89 (Figure 5A–D).

### 2.3. I_Kr_ Is Not Affected by β-Adrenergic Receptor Stimulation, PKA Inhibition, or CaMKII Inhibition

The consequences of KN-93 and H-89 treatment, as well as the effects of ISO in the absence and presence of these inhibitors, were studied on I_Kr_ under similar experimental conditions. Neither ISO, nor the inhibitors applied alone or in combination with ISO, modified the profile (Figure 6A), the mid-plateau density (Figure 6B), the peak density (Figure 6C), and the integral (Figure 6D) of I_Kr_.

## 3. Discussion

In the present study, the effect of 10 nM ISO, corresponding to a moderate sympathetic activation of the heart, was studied on two K^+^ currents governing terminal repolarization: I_K1_ and I_Kr_. I_K1_ was increased by ISO in the plateau phase of the canine ventricular AP, while I_Kr_ was not influenced by ISO. This is the first study to demonstrate that the β-adrenergic-pathway-dependent augmentation of the canine I_K1_ is not mediated by the PKA pathway, since the ISO effect was not prevented by pretreatment with the PKA inhibitor H-89. In contrast, the CaMKII inhibitor KN-93 abolished the effect of ISO on I_K1_, indicating that CaMKII is responsible for the effect. No synergy was observed in the contribution of CaMKII and PKA, since the I_K1_ densities measured in the presence of ISO + KN-93 and ISO + KN-93 + H-89 were not different. Also, the baseline activity of CaMKII or PKA is not involved in the regulation of I_K1_, since neither KN-93 alone nor H-89 alone modulated the density of I_K1_ under baseline conditions.

To explain our findings, there are two plausible possibilities based on experimental data in the literature. ß-adrenergic activation increases CaMKII activity in guinea pig cardiomyocytes through a nitric oxide (NO)-dependent, but cAMP-independent, pathway [26]. Based on this, the authors concluded that the ß-adrenergic receptor stimulation may activate CaMKII by a novel direct pathway involving nitrogen oxide synthase. Indeed, NO increased CaMKII activity in human atrial myocytes and K_ir_2.1 currents [27], and nitrosylation of CaMKII-delta has been shown to mediate the effect of 100 nM ISO in murine cardiac cells [28]. The involvement of the guanine nucleotide exchange protein activated by cAMP (EPAC) in the stimulation of CaMKII has also been reported [29,30,31]. It has been suggested that the NO-dependent activation of CaMKII is mediated by an EPAC-related pathway in murine and rabbit myocytes via a cAMP → EPAC → NO → CaMKII sequence [32]. Alternatively, the activation of CaMKII may be mediated by a cAMP→ EPAC → Rap → PLC-epsilon sequence [30]. Furthermore, the contribution of ß-arrestins and related signalosomes cannot be excluded either [33]. Further studies are required to identify the exact signal transduction mechanism(s) in canine ventricular cells mediating ß-adrenergic regulation of I_K1_.

It was unexpected to find that ISO only increased I_K1_ in the plateau voltage range of the AP but not at more negative voltages, such as during the terminal repolarization, when the current density is maximal. This suggests that the maximal overall K^+^ conductance was unaffected by ß-adrenergic activation. It is likely that the number of active Kir2.1 channels and their maximal conductance remained unchanged, although two opposing effects — such as an increased conductance but decreased Kir2.x channel expression — could also explain this observation. Our results suggest that CaMKII-induced phosphorylation may reduce the inward-going rectification of I_K1_, resulting in a larger outward I_K1_ current at positive membrane potentials, as seen in our experiments. A similar phosphorylation-dependent alteration in the rectification of Kir2.1/KCNJ2 channels has been reported by Kalscheur et al. in an R67Q mutant channel [5]. In wild-type Kir2.1 channels, the phosphorylation of the channel on Ser425 increases I_K1_, an effect which is lost in the R67Q mutants. The authors attributed this to an increased I_K1_ “rectification index” [5]. As the AP duration is most sensitive to changes in the net membrane current when the repolarization is slow [34], such as during the AP plateau phase, even a small increase in I_K1_ under the AP plateau can significantly shorten the AP.

In agreement with our results, Nagy et al. [20] also reported a CaMKII-mediated [Ca^2+^]_i_-dependent I_K1_ augmentation in a wide range of membrane voltages in canine ventricular cells. However, there is an apparent contradiction between the results of the two studies regarding the peak I_K1_ density. Nagy et al. reported a significantly larger peak I_K1_ density under APVC conditions in case of larger [Ca^2+^]_i_ (attributed to CaMKII activation), whereas we found no significant effect of ISO on the peak I_K1_ density. These contradictory results may originate from the slightly different experimental conditions. In their canine APVC studies, Nagy et al. set the [Ca^2+^] of the pipette solutions with BAPTA and CaCl_2_ to low (~160 nM) and high (~900 nM) levels to elucidate the [Ca^2+^]_i_ dependency of I_K1_, whereas we used adrenergic stimulation to achieve CaMKII activation, with no Ca^2+^ buffers applied.

In rabbit ventricular cells, under experimental conditions quite similar to ours, Hegyi et al. [35] found that the inhibition of CaMKII did not alter I_K1_ parameters, and the application of 10 nM ISO did not affect the peak I_K1_ density when intracellular calcium homeostasis was preserved, results which are in good agreement with our data (Figure 2B and Figure 5). In the study by Hegyi et al., however, 10 nM ISO increased the I_K1_ net charge significantly (by 23%) compared to control conditions. The same comparison did not reach statistical significance in our dataset, although 10 nM ISO-treated cells had an approximately 16% (11 mC/F) larger I_K1_ integral compared to the control cells (Figure 2C). The authors [35] did not specifically comment on which part of the I_K1_ current trace became larger to cause the significant increase in the I_K1_ integral. Since the peak I_K1_ did not change, the most plausible explanation is a significant increase in I_K1_ under the plateau phase, just like what we found in the present study. When Ca^2+^_i_ was buffered with BAPTA, the authors found no significant differences in either the I_K1_ peak or I_K1_ net charge between basal conditions and ß-adrenergic stimulation with 10 nM ISO. In our study, KN-93 had a similar effect, preventing the ISO-induced increase in I_K1_ under the AP plateau (Figure 3B,C and Figure 4B,C). Furthermore, cells in the control and KN-93 + ISO groups had similar I_K1_ integrals (66.7 ± 4.4 mC/F and 63.2 ± 8.7 mC/F, respectively) that were considerably (respectively, around 16% and 23%) smaller than in the ISO-treated cells (77.7 ± 8.1 mC/F), but these differences in the net charge carried by I_K1_ did not reach statistical significance. The observations of both studies can be explained if we accept that ß-adrenergic receptor activation increases CaMKII activity, and in turn, CaMKII upregulates the I_K1_ current. Therefore, if CaMKII activation is prevented either by Ca^2+^_i_ buffering, or by direct CaMKII inhibition, I_K1_ upregulation is abolished.

It is well established that I_K1_ is substantially activated by phosphatidylinositol 4,5-bisphosphate (PIP_2_) [36,37], and that β-adrenergic receptor activation leads to increased PIP_2_ levels [38]. In the latter article by Xu et al., the authors suggest that the β-adrenergic receptor activation raises the PIP_2_ levels because of the PKA-dependent phosphorylation and subsequent activation of phosphatidylinositol-4-phosphate 5-kinase gamma (PIP5Kγ). However, Xu et al. applied *30 µM H-89 extracellularly* to inhibit PKA. Notably, H-89 is a potent and selective inhibitor of β_1_ (and β_2_) adrenergic receptor ligand binding with an approximate K_i_ of 350 nM (and 180 nM) on human airway smooth muscle cells [39]. This β-receptor inhibitor property must be considered when applying H-89 extracellularly. Therefore, the results that Xu et al. show in Figure 4B [38] can be attributed to *β-adrenergic receptor inhibition by H-89*. This was also the reason why we used a lower concentration of H-89 (3 µM) *in the pipette solution*. Considering all of these, an alternative explanation for our findings is possible, besides the suggested CaMKII-dependent phosphorylation of Kir channels. As β-adrenergic receptor activation increases PIP_2_ levels [38], it could also enhance I_K1_ through this mechanism. Since in our experiments, intracellularly applied KN-93 prevented the augmentation of I_K1_, but H-89 did not have such an effect, this putative mechanism is clearly not mediated by PKA, and CaMKII might play a role in it.

In contrast to our findings on I_K1_, and previous studies on both I_Ks_ [40,41] and I_K1_ [1,2,3], the I_Kr_ density was not altered by ß-adrenergic activation in our study, nor was it sensitive to the inhibition of either CaMKII or PKA under our experimental conditions. In rabbit ventricular cells under APVC conditions and in normal intracellular calcium homeostasis, similar to our experimental settings, CaMKII inhibition did not alter I_Kr_ parameters either, whereas ß-adrenergic stimulation with 10 nM ISO caused an 8.1% increase in the peak I_Kr_ density and a 23.1% increase in the current integral [35]. In guinea pig myocytes, I_Kr_ was increased by PKC activation [12] but suppressed by PKA activation [13]. In contrast, PKA activation increased the I_Kr_ amplitude in canine myocytes [11], while it decreased it in human and rat myocytes [8,9]. Since both the PKA- and PKC-related pathways are [Ca^2+^]_i_-dependent, it is not surprising that these effects were strongly influenced by the [Ca^2+^]_i_ levels [12]. Similarly, ISO increased the I_Kr_ density when [Ca^2+^]_i_ was kept low ([Ca^2+^]_i_ was buffered by EGTA, and Ca^2+^ entry was blocked by nifedipine) in conventional voltage clamp experiments [11] but was not altered in the present study when the Ca^2+^ homeostasis was intact and experiments were carried out in APVC conditions. These findings together suggest that ß-adrenergic stimulation might modify I_Kr_ in the canine heart, but likely through two opposing pathways, which may cancel each other out under intact intracellular Ca^2+^ homeostasis. Further investigations are necessary to clarify the exact role of these mechanisms.

## 4. Materials and Methods

### 4.1. Animals

Adult mongrel dogs of both sexes were anesthetized using intramuscular injections of 10 mg/kg ketamine hydrochloride (Calypsol, Richter Gedeon, Budapest, Hungary) +1 mg/kg xylazine hydrochloride (Sedaxylan, Eurovet Animal Health BV, Bladel, The Netherlands) according to the protocol approved by the local Animal Care Committee of the University of Debrecen (license N°: 9/2015/DEMÁB). All animal procedures conform to the guidelines laid down in the Declaration of Helsinki in 1964 and its later amendments, and to the Guide to the Care and Use of Experimental Animals (Vol. 1, 2nd ed., 1993, and Vol. 2, 1984, Canadian Council on Animal Care).

### 4.2. Isolation of Cardiomyocytes

Single canine myocytes were obtained by enzymatic dispersion using the segment perfusion technique, as previously described [34,42]. A wedge-shaped section of the ventricular wall, supplied by the left anterior descending coronary artery, was cannulated and perfused with a nominally Ca^2+^-free Joklik solution (Minimum Essential Medium Eagle, Joklik Modification). After washing out the blood (5 min) using this solution, it was supplemented with 1 mg/mL collagenase (Type II, Worthington Biochemical Co., Lakewood, NJ, USA; representing final activity of 224 U/mL) and 0.2% bovine serum albumin (Fraction V.) containing 50 µM Ca^2+^. This perfusion typically lasted for 30 min. Following this period, the tissue was minced, and the cells were released by gentle agitation. Finally, the normal external Ca^2+^ concentration was gradually restored. The cells were stored in Minimum Essential Medium Eagle at 15 °C before use. This protocol yielded myocytes showing clear cross-striations, dominantly of a midmyocardial origin.

### 4.3. Electrophysiology

Cells were superfused in a plexiglass chamber under an inverted microscope with a modified Tyrode solution by gravity flow at a rate of 1–2 mL/min. This superfusate contained NaCl, 121; KCl, 4; CaCl_2_, 1.3; MgCl_2_, 1; HEPES, 10; NaHCO_3_, 25; and glucose, 10 (mM) at pH = 7.35 with an osmolarity of 300 mOsm. The bath temperature was set to 37 °C using a temperature controller (Cell MicroControls, Norfolk, VA, USA). Electrical signals were amplified and recorded (MultiClamp 700A or 700B, Molecular Devices, Sunnyvale, CA, USA) under the control of a pClamp software (version 10.6, Molecular Devices) following analog–digital conversion (Digidata 1440A or 1332, Molecular Devices). Electrodes with tip resistances of 2–3 MΩ when filled with pipette solution were made from borosilicate glass. The pipette solution contained K-aspartate, 120; KCl, 30; KOH, 10; MgATP, 3; HEPES, 10; Na_2_-phosphocreatine, 3; EGTA, 0.01; and cAMP, 0.002 at pH = 7.3 and osmolality of 285 mOsm. The series resistance was kept between 4 and 8 MΩ. The experiment was discarded when the series resistance changed substantially during the measurement. After establishing the whole-cell configuration, cells were continuously paced in current-clamp mode for at least 10 min at a cycle length of 0.7 s, corresponding to the normal heart rate of dogs. See further details in [25,43].

The experiments were performed using the APVC technique according to the method previously described [24,44,45]. To avoid the consequences of cell-to-cell variations in AP configurations, measurements were performed using a “canonic” AP as a command signal instead of the own AP of the cell. This canonic AP was a representative midmyocardial canine AP with average parameters. Cells were paced with the “canonic” AP at a cycle length of 0.7 s. The application of uniform command APs allowed for the comparison of the individual current traces. I_K1_ was dissected by 50 µM BaCl_2_, applied after pretreatment with 1 µM E-4031 plus 0.5 µM HMR-1556 to block I_Kr_ and I_Ks_, respectively. When measuring I_Kr_ as a 1 µM E-4031 sensitive current, the cells were pretreated with 0.5 µM HMR-1556 to block I_Ks_. Before and after 5 min of superfusion with the inhibitor (BaCl_2_ for I_K1_ or E-4031 for I_Kr_), 20 consecutive current traces were recorded and averaged in order to reduce noise and trace-to-trace fluctuations. The drug-sensitive current was obtained by subtracting the averaged post-drug current trace from the averaged pre-drug one. These dissected currents were evaluated by determining their maximal amplitudes (peak currents), their amplitudes measured at the half-duration of the command AP (mid-plateau current amplitudes), and the total charge carried by the current (current integrals). All these parameters were normalized to cell capacitance, which was calculated based on hyperpolarizing the cells from +10 mV to −10 mV for 15 ms.

Conventional voltage clamp experiments of I_K1_ (shown in Figure 4) were performed with the same pipette solution as the APVC measurements. For these experiments, the bath solution was supplemented with 1 µM E-4031 and 0.5 µM HMR-1556 for I_Kr_ and I_Ks_ blockade, respectively. The steady-state I_K1_ was defined as the BaCl_2_-sensitive current. To obtain current traces, 250 ms long voltage test pulses were delivered in 10 mV increments from the holding potential of −80 mV at a rate of 1 Hz. After applying 50 µM BaCl_2_ for 5 min, the recording was repeated, and these current traces were subtracted from their respective pre-BaCl_2_ ones. BaCl_2_-sensitive current readings from the last 5 ms of the test pulses were averaged, normalized to the cell capacitance, plotted against their respective test potentials, and regarded as the I_K1_ current densities.

### 4.4. Chemicals

Chemicals were purchased from Sigma-Aldrich (St. Louis, MO, USA), except for HMR-1556, KN-93, and H-89, which were purchased from Tocris Bioscience (Bristol, UK). An ISO stock solution of 10 µM was freshly created right before each experiment, and this was diluted to a 10 nM final concentration in the bath solution. ISO application started 5 min before the start of APVC experiments, and from that time point, it was continuously perfused in the bath. Besides ISO, the potassium channel inhibitors HMR-1556, E-4031, and BaCl_2_ were also used in the bath solution.

When applied extracellularly, 1 µM KN-93 has been shown to abolish I_Kr_ in rabbit and guinea pig ventricular myocytes [46], whereas H-89 is shown to be a potent and selective inhibitor of β_1_ and β_2_ adrenergic receptor ligand binding with respective K_i_ values of around 350 nM and 180 nM on human airway smooth muscle cells [39]. Therefore, both kinase inhibitors (1 µM KN-93 and 3 µM H-89) were applied in the pipette solution.

### 4.5. Statistics

Results are expressed as mean ± SEM values, and “*n*” denotes the number of myocytes studied. The experimental groups were statistically compared with one-way ANOVA, followed by Dunnett’s post hoc test if the ANOVA yielded significant results. In the results presented in Figure 2, Figure 3 and Figure 4, the 10 nM ISO-treated group served as the reference group, whereas in the comparisons shown in Figure 5 and Figure 6, all other groups were compared to the untreated “control” group in Dunnett’s post hoc test. Differences were considered statistically significant when *p* was less than 0.05.

## Figures and Tables

**Figure 1 ijms-25-11609-f001:**
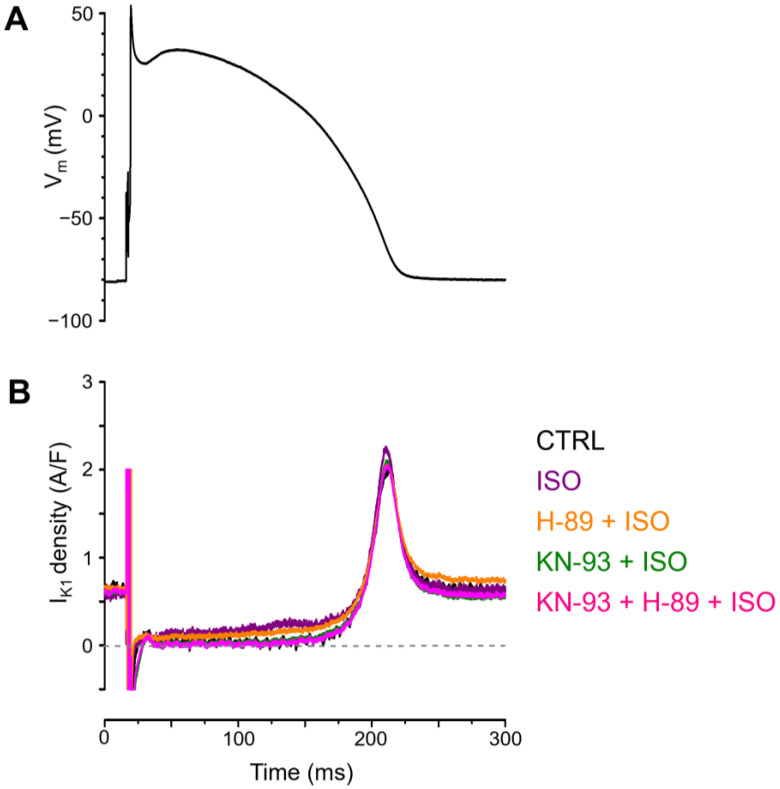
(**A**) Command action potential and (**B**) representative I_K1_ current traces obtained under APVC conditions in untreated control cells (CTRL), with ß-adrenergic stimulation in the presence of 10 nM ISO (ISO), ß-adrenergic stimulation following CaMKII inhibition with 1 µM KN-93 (KN-93 + ISO), ß-adrenergic stimulation following PKA inhibition with 3 µM H-89 (H-89 + ISO), and ß-adrenergic stimulation with the inhibition of both kinases (KN-93 + H-89 + ISO).

**Figure 2 ijms-25-11609-f002:**
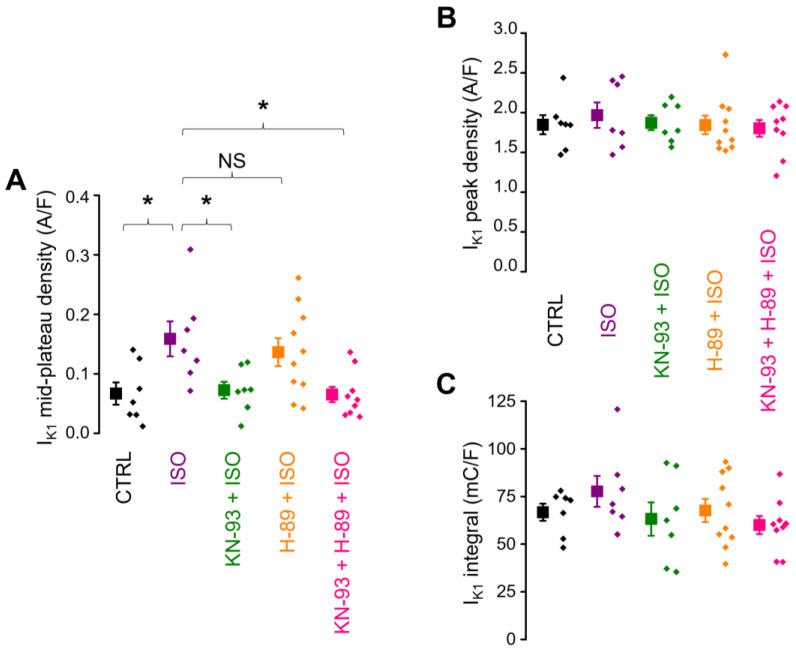
The effect of ß-adrenergic stimulation on I_K1_ and the suppressibility of this effect by CaMKII inhibition (KN-93) and PKA inhibition (H-89). Data obtained in 5 groups of myocytes are compared: untreated control (CTRL; *n* = 7/5), ß-adrenergic stimulation alone (ISO; *n* = 7/6), ß-adrenergic stimulation following CaMKII inhibition (KN-93 + ISO; *n* = 7/6), ß-adrenergic stimulation following PKA inhibition (H-89 + ISO; *n* = 10/7), and ß-adrenergic stimulation with the inhibition of both kinases (KN-93 + H-89 + ISO; *n* = 9/7). The “*n*” numbers in parentheses indicate the number of cells over the number of animals used in that experimental group. (**A**) Mid-plateau current densities, (**B**) peak current densities, (**C**) current integrals. Symbols and bars are mean ± SEM, and small dots represent individual data. Asterisks (*) indicate significant differences (*p* < 0.05) between groups. NS: not significant.

**Figure 3 ijms-25-11609-f003:**
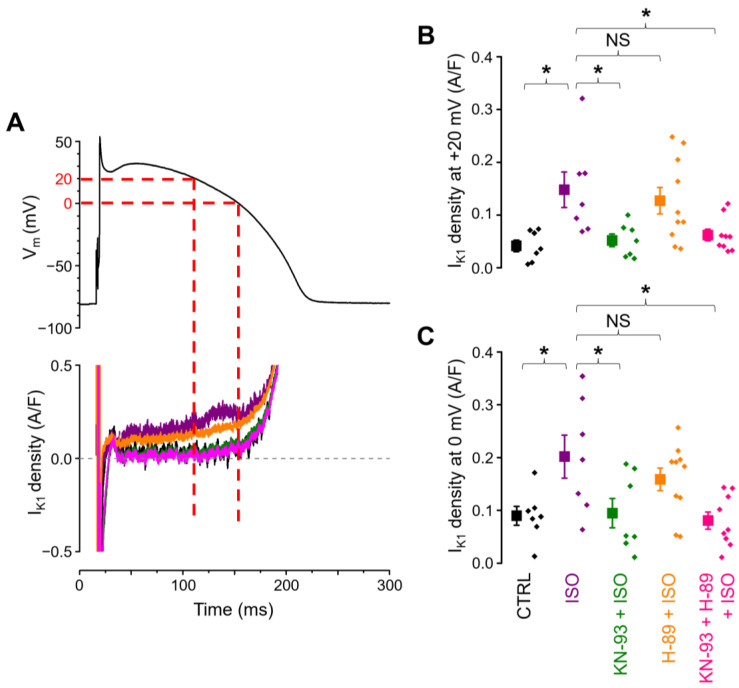
(**A**) Command action potential (above) and representative enlarged I_K1_ current traces flowing during the action potential plateau (below). Dashed lines indicate the corresponding voltage and current values at 0 mV and at +20 mV. (**B**,**C**) I_K1_ densities measured at +20 mV and at 0 mV, respectively, in the 5 cell groups of control (CTRL; *n* = 7/5), ß-adrenergic stimulation alone (ISO; *n* = 7/6), ß-adrenergic stimulation following CaMKII inhibition (KN-93 + ISO; *n* = 7/6), ß-adrenergic stimulation following PKA inhibition (H-89 + ISO; *n* = 10/7), and ß-adrenergic stimulation with the inhibition of both kinases (KN-93 + H-89 + ISO; *n* = 9/7). The “*n*” numbers in parentheses indicate the number of cells over the number of animals used in that experimental group. Symbols and bars are mean ± SEM, and small dots represent individual data. Asterisks (*) indicate significant differences (*p* < 0.05) between groups. NS: not significant.

**Figure 4 ijms-25-11609-f004:**
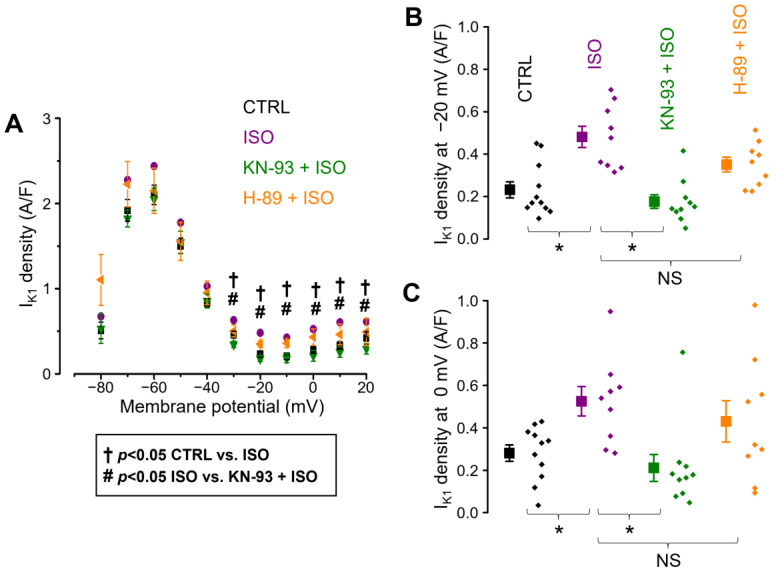
The effects of ISO alone and in the presence of KN-93 and H-89 on the I_K1_ density under conventional voltage clamp conditions. (**A**) I_K1_ densities measured at the end of various test potentials (shown on the abscissa) lasting for 250 ms and rising from the holding potential of −80 mV in the 4 cell groups of control (CTRL; *n* = 11/3), ß-adrenergic stimulation alone (ISO; *n* = 9/5), ß-adrenergic stimulation following CaMKII inhibition (KN-93 + ISO; *n* = 10/3), and ß-adrenergic stimulation following PKA inhibition (H-89 + ISO; *n* = 9/5). The “n” numbers in parentheses indicate the number of cells over the number of animals used in that experimental group. Detailed statistical analyses of the I_K1_ densities measured at −20 mV and 0 mV are depicted in panels (**B**) and (**C**), respectively. Symbols and bars are mean ± SEM, and small dots represent individual data. On panel (**A**)**,** daggers (†) and hash signs (#) indicate significant differences (*p* < 0.05) between the control and ISO groups and between the ISO and KN-93 + ISO groups, respectively. On panels (**B**,**C**), asterisks (*) indicate significant differences (*p* < 0.05) between groups. NS: not significant.

**Figure 5 ijms-25-11609-f005:**
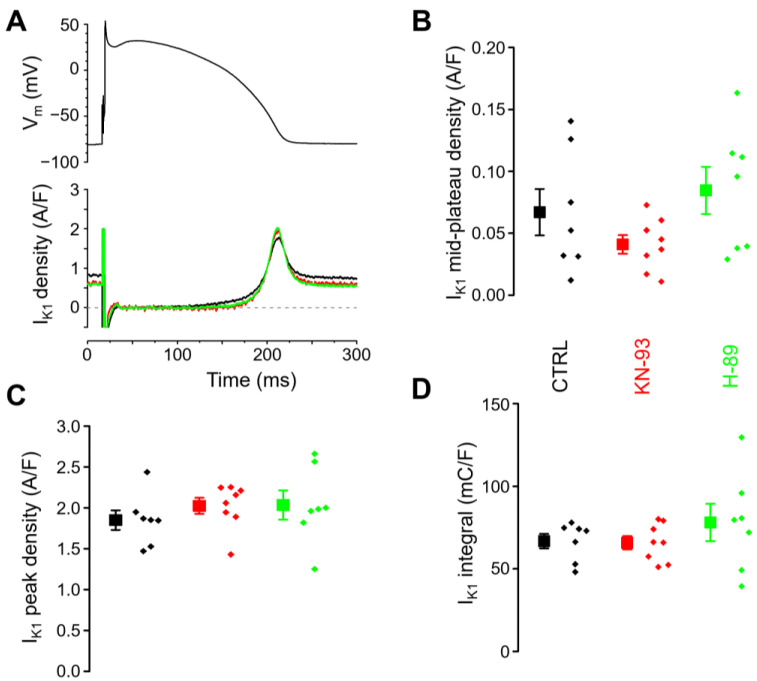
CaMKII inhibition with KN-93 or PKA inhibition with H-89 has no effect on I_K1_ under baseline conditions. The cell groups of control (CTRL; *n* = 7/5), CaMKII inhibition (KN-93; *n* = 8/5), and PKA inhibition (H-89; *n* = 7/4) were compared. The “*n*” numbers in parentheses indicate the number of cells over the number of animals used in that experimental group. (**A**) Command action potential (above) and the corresponding representative I_K1_ current traces flowing during the action potential (below). (**B**) Mid-plateau current densities, (**C**) peak current densities, and (**D**) current integrals. Symbols and bars are mean ± SEM, and small dots represent individual data. The arithmetic means of the groups were not significantly different in any parameters with one-way ANOVA.

**Figure 6 ijms-25-11609-f006:**
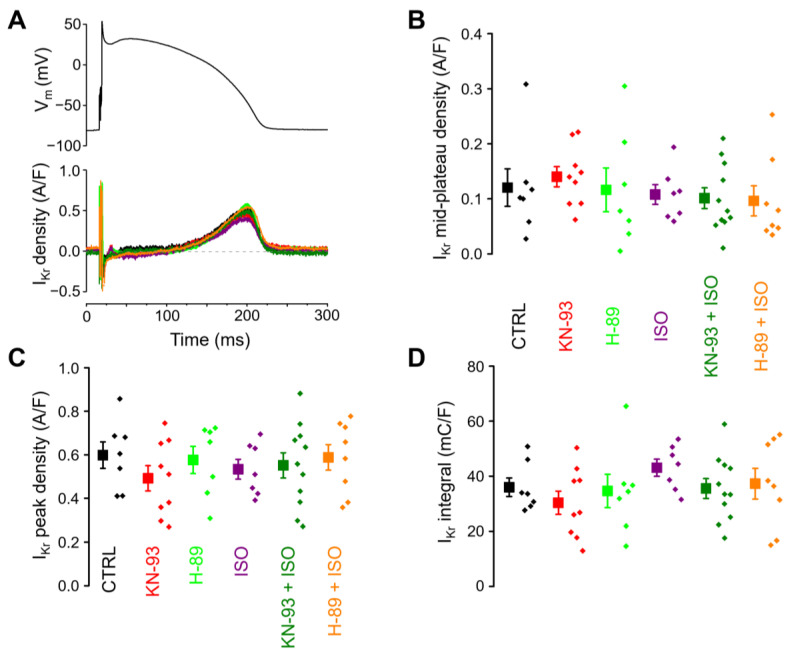
The effects of PKA and CaMKII inhibition with or without ß -adrenergic receptor activation on I_Kr_. The cell groups of control (CTRL; *n* = 7/5), CaMKII inhibition under baseline conditions (KN-93; *n* = 9/5), PKA inhibition under baseline conditions (H-89; *n* = 7/4), ß-adrenergic stimulation alone (ISO; *n* = 7/6), ß-adrenergic stimulation following CaMKII inhibition (KN-93 + ISO; *n* = 11/6), and ß-adrenergic stimulation following PKA inhibition (H-89 + ISO; *n* = 8/6). The “n” numbers in parentheses indicate the number of cells over the number of animals used in that experimental group. (**A**) Command action potential (above) and the corresponding representative I_Kr_ current profiles flowing during the action potential (below). (**B**) Mid-plateau current densities, (**C**) peak current densities, and (**D**) current integrals. Symbols and bars are mean ± SEM, and small dots represent individual data. The arithmetic means of the groups were not significantly different in any parameters with one-way ANOVA.

## Data Availability

Data are available upon reasonable request. Inquiries regarding research data should be directed to the corresponding author.

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
