# Peer review of "Beta-Adrenergic Activation of the Inward Rectifier K+ Current Is Mediated by the CaMKII Pathway in Canine Ventricular Cardiomyocytes"

_ijms, 2024, doi:10.3390/ijms252111609_

Round 1
Reviewer 1 Report
Comments and Suggestions for Authors
Reviewing the review manuscript entitled, “Beta-adrenergic activation of the inward rectifier K+ current is mediated by the CaMKII pathway in canine ventricular cardiomyocytes” by Kovács Z et al., this is an article focusing on the inward rectifier K+ current kinetics of beta-adrenergic activation mediated by the CaMKII pathway in cardiomyocytes. This is a new finding regarding catecholamine-induced inward K+ currents and is extremely interesting. The authors need to respond to the following concerns for reaching to an acceptable quality.
There is no particular problem with the experimental system for performing in vitro experiments, such as isolating cardiomyocytes. However, isolating cardiomyocytes requires skilled techniques. Who was in charge of this? Please include this in the authors' response.
The authors should modify Figure 1. It is difficult to distinguish between each experiment.
What does the increase in density of Ik1 plateau phase by ISO mean clinically? Is it a significant shortening of AP as the authors describe? It is generally believed that Ik1 function depends on the plasma K+ concentration, but how do catecholamines and K+ concentrations control Ik1 function?
Author Response
- There is no particular problem with the experimental system for performing in vitro experiments, such as isolating cardiomyocytes. However, isolating cardiomyocytes requires skilled techniques. Who was in charge of this? Please include this in the authors' response.
Cardiomyocyte isolation is a very delicate technique which must be well established and continuously monitored to be reliable. Our laboratory has a long history of successfully isolating primary cardiomyocytes, dating back to the 1990’s. There are two critical elements for successful cell isolation. First, the extraction and cannulation of the heart needs to be as fast as possible, to avoid ischaemic damage of the cells. The second one is to find the appropriate moment to harvest the cells during the enzymatic isolation procedure.
The cells used for this study were routinely isolated by PhD students in our laboratory, mainly by Zsigmond Kovács, Csaba Dienes, Dénes Kiss and Tamás HézsÅ‘. The cell isolation procedure is always supervised by a senior colleague, either Norbert Szentandrássy, or Balázs Horváth. Our junior colleagues can start doing cell isolation on their own after they take part in the cell isolation procedure for at least 4-6 months to learn the technique and gain the necessary skills.
- The authors should modify Figure 1. It is difficult to distinguish between each experiment.
We agree with the reviewer that it is hard to distinguish between the experiments of CTRL, KN-93+ISO and the KN-93+H-89+ISO traces under the plateau phase, and the IK1 peak under CTRL, KN-93+ISO, H-89+ISO and KN-93+H-89+ISO conditions on Figure 1. We created the figure this way intentionally, to demonstrate the statistical average of our experiments. As it is clear from Figure 2, there were no differences between the groups with regards to their average IK1 peak density. Furthermore, as both Figure 2 and Figure3 demonstrate, no differences were observed between the groups of CTRL, KN-93+ISO, and KN-93+H-89+ISO with regards to the IK1 current densities under the plateau phase. This is why all the authors agreed to present the representative current traces in exactly this way to faithfully illustrate the average values.
- What does the increase in density of Ik1 plateau phase by ISO mean clinically? Is it a significant shortening of AP as the authors describe? It is generally believed that Ik1 function depends on the plasma K+ concentration, but how do catecholamines and K+ concentrations control Ik1 function?
ISO is a β-adrenergic agonist, and its effects on the cardiomyocyte potassium currents are well studied. The shape of the plateau phase of the AP depends on the fine balance between inward (depolarizing) and outward (repolarizing) currents flowing under this phase [doi: 10.1007/s00424-014-1465-7]. As the membrane impedance of the cardiomyocytes are large under the plateau [DOI: 10.1038/188495b0, doi: 10.1113/jphysiol.1962.sp006849] even a small change in this fine balance might result in a significant change in the AP length [doi: 10.1113/jphysiol.2014.279554].
The increased density of the IK1 during the AP plateau shifts this balance towards the outward currents, therefore inducing an earlier repolarization of the cardiomyocyte membrane, resulting in a shorter AP. The clinical result of this is a shorter ventricular AP when the heart rate is elevated under physiological conditions. Under beta receptor activation, the major repolarizing current will be IKs, termed the “repolarization reserve” [DOI: 10.1111/j.1540-8159.1998.tb00148.x]. Although it is very hard to calculate or predict how much AP shortening (or QT shortening in a clinical setting) is due to IK1 under a certain sympathetic stimulus, it must be noted that under beta-adrenergic stimulation IK1 contributes significantly to ventricular repolarization [doi: 10.1007/s00424-014-1465-7].
Cathecolamines control the ionic currents of the heart muscle cells through β1-receptor signalization. The β1-receptor is a G-protein-coupled receptor. It uses the Gs protein to increase adenylyl cyclase activity, and through several pathways (i.e. EPAC, PKA), modifies several ionic currents. To find out, which of these pathways might control the IK1 current was one aim of our study. Regarding extracellular K+, the channel conductance of Kir channels is proportional to the square root of extracellular K+ concentration, but Na+ ions also have a facilitatory effect on the single-channel conductance [DOI: 10.1007/bf01870774; DOI: 10.1016/j.bbamem.2011.02.016]. Therefore, elevating extracellular K+ on one hand increases single-channel conductance, but on the other hand, the electrochemical gradient of K+ decreases, decreasing the driving force for IK1 current. An article about Weiss et al. [doi: 10.1161/CIRCEP.116.004667] provides a detailed description of the multifaceted nature of the electrophysiological changes in hypo-, and hyperkalaemia.
Thinking about these questions asked by the Reviewer also made the authors reconsider and supplement the possible explanations of the experimental results regarding IK1.
It is well established that IK1 is substantially activated by phosphatidylinositol 4,5-bisphosphate (PIP2) [DOI: 10.1126/science.273.5277.956, DOI: 10.1126/stke.2001.111.re19], and that β-adrenergic receptor activation leads to in-creased PIP2 levels [doi: 10.1074/jbc.M113.527952]. In this latter article by Xu et al., authors suggest that the β-adrenergic receptor activation raises the PIP2 levels because of the PKA-dependent phosphorylation and therefore activation of phosphatidylinositol-4-phosphate 5-kinase gamma (PIP5Kγ). However, Xu et al., applied 30 µM H-89 extracellularly to inhibit PKA. Notably, H-89 is a potent and selective inhibitor of β1 (and β2) adrenergic receptor ligand binding with an approximate Ki of 350 nM (and 180 nM) on human airway smooth muscle cells [PMID: 9918542]. This β-receptor inhibitor property must be considered when applying H-89 extracellularly. Therefore, the results that Xu et al. show in Figure 4. B [doi: 10.1074/jbc.M113.527952] can be attributed to β-adrenergic receptor inhibition by H-89. This was also the reason why we used a lower, 3 µM H-89 in the pipette solution. Considering all of these, an alternative explanation for our findings is possible, besides the suggested CaMKII-dependent phosphorylation of Kir channels. As β-adrenergic receptor activation increases PIP2 levels [doi: 10.1074/jbc.M113.527952], it could also enhance IK1 through this mechanism. Since in our experiments, intracellularly applied KN-93 prevented the augmentation of IK1, but H-89 did not have such an effect, this putative mechanism is clearly not mediated by PKA, and CaMKII might play a role in it.
This section was added to the Discussion of the article as well (lines 280-296).
Reviewer 2 Report
Comments and Suggestions for Authors
The aim of this study was to examine the mechanisms of β-adrenergic receptor stimulation by isoproterenol (ISO) on the inward rectifier (IK1) and the rapid delayed rectifier (IKr) K+ currents in isolated canine left ventricular cardiomyocytes. Channel activities were measured by action potential voltage clamp (APVC) and conventional voltage clamp techniques. Activities of the specific channels were identified by using the specific inhibitors, BaCl2 and E-4031, respectively. Isoproterenol significantly increased IK1 under the plateau phase of the action potential. Isoproterenol did not affect the peak current density flowing during terminal repolarization, or the overall IK1 integral. The ISO-induced enhancement of IK1 was blocked by the calcium/calmodulin kinase II inhibitor KN-93. In contrast, isoproterenol had no effect on IKr.
The topic and the results are of interest and the manuscript is well-written. Nevertheless, there are also some concerns to be addressed.
1. Line 41, please correct “beta-adrenergic receptor”.
2. Lines 42-43, in which sense IK1 and IKr are involved in terminal repolarization? In which phase of repolarization these channels are active regarding, for example, IKs, which was mentioned earlier.
3. Lines 55-56, it is stated that isoproterenol has been demonstrated to stimulate Ikr in canine cardiomyocytes; what is the mechanism of the discrepancy between cited study and the present study regarding this effect?
4. How many dogs were used in the experiments? How the animals were sacrificed after tissue sampling?
5. Did you discriminate between cardiomyocytes obtained from male and female dogs? Do you expect sex-dependent differences?
6. Lines 177-178, in which sense the concentration of ISO used in this study “corresponds to physiological stimulation of the heart”?
7. Was the effect of ISO on PKA activity measured? Has forskolin any effect on IK1 in this model?
8. The significant limitation of the present study is that the role of PKA and CaCMK was assessed using only single inhibitors of these kinases. Are these inhibitors, at the doses applied, sufficiently specific to discriminate between them?
Author Response
- Line 41, please correct “beta-adrenergic receptor”.
We are not sure what the reviewer meant on this correction. We checked the main text of our submitted article, and there were only two instances where “beta” was used, everywhere else we used the Greek letter “β” consistently throughout the main text (12 occurrences). We corrected “beta” to “β” in line 85 and line 89, to be completely consistent. The only exception is the title, because in our experience, search engines and citation manager programs do not handle Greek letters very well.
We also corrected “ß-adrenoceptor” to “ß-adrenergic receptor” in line 45 and 207.
- Lines 42-43, in which sense IK1 and IKr are involved in terminal repolarization? In which phase of repolarization these channels are active regarding, for example, IKs, which was mentioned earlier.
In case of IKr, our previous study [doi: 10.1111/j.1748-1716.2007.01674.x] shows that it is active during the whole repolarization, however it reaches its highest current density a few milliseconds before the maximal rate of repolarization of the AP (the terminal repolarization phase), so we can say that IKr responsible for starting the terminal repolarization. We have also found that the amplitude of IK1 slowly rising during the AP plateau and reaches its peak density 1 ms after the maximal rate of repolarization of the AP, therefore it has a role in the ending of the AP, and keeping the resting membrane potential stable. See also Figure 4. of the extensive review of Varro et al. [doi: 10.1152/physrev.00024.2019] on this topic.
Without PKA-dependent stimulation, IKs is almost negligible under the AP. However, when the sympathetic nervous system is activated, IKs becomes the dominant repolarizing current under the AP [doi: 10.1007/s00424-014-1465-7].
- Lines 55-56, it is stated that isoproterenol has been demonstrated to stimulate Ikr in canine cardiomyocytes; what is the mechanism of the discrepancy between cited study and the present study regarding this effect?
In the experiments of Harmati et al. [doi: 10.1111/j.1476-5381.2010.01092.x], IKr was measured with conventional voltage-clamp, where the current was activated by 250-ms-long depolarizing pulses to +10 mV applied at a rate of 0.05 Hz, followed by repolarization to -40 mV. IKr tail current amplitudes were determined as the difference between the tail current peak (occurring immediately after setting the membrane potential to -40 mV) and the steady-state current value observed at the end of the repolarizing step. Also, in the cited study by Harmati et al. [Ca2+]i was buffered by EGTA and Ca2+ entry was blocked by nifedipine. Under these conditions, 10 nM ISO increased IKr tail amplitudes only by approximately 15 %. In our experiments, IKr was measured with action potential voltage-clamp, and we found no significant differences between the cell groups studied. The discrepancy in the observed effect between the study of Harmati et al. and the current study is most likely due to the difference in the applied command voltage pulse.
A shorter version of this answer was also inserted into the Discussion section (lines 308-311):
“Similarly, ISO increased IKr density when [Ca2+]i was kept low ([Ca2+]i was buffered by EG-TA and Ca2+ entry was blocked by nifedipine) in conventional voltage-clamp experiments [11], but was not altered in the present study when the Ca2+ homeostasis was intact and experiments were done in APVC conditions.”
- How many dogs were used in the experiments? How the animals were sacrificed after tissue sampling?
We sacrificed 36 animals to obtain our results in the submitted article. For the cell isolation, the animals are anesthetized, and after reaching deep anesthesia (judged by the absence of the cornea reflex and the paw withdrawal reflex), their hearts were extracted from the thoracic cavity.
To better illustrate the usage of the model animals, the number of animals used for each experimental group are also mentioned in the figure legends.
- Did you discriminate between cardiomyocytes obtained from male and female dogs? Do you expect sex-dependent differences?
For this paper we did not discriminate between the male and female dogs. Based on data from literature, we could expect differences between males and females, as there are sex-dependent differences in average heart rate [doi: 10.56808/2985-1130.2337], and blood pressure [http://dx.doi.org/10.1186/2042-6410-3-7] observed in dogs. However, we still do not know if those are created by the difference in cardiomyocyte electrophysiology, or a difference in the vegetative nervous systems control of the heart, created by the sex-hormones. We are currently analyzing our cellular electrophysiology data from the last two decades to see if there are any electrophysiological differences between the two genders. There are no sex-dependent differences regarding IKr and IK1 with action potential voltage clamp under control conditions.
- Lines 177-178, in which sense the concentration of ISO used in this study “corresponds to physiological stimulation of the heart”?
In most of the studies tackling beta-adrenergic receptor stimulation, the agonists are usually used in a concentration that saturates these receptors, e.g. ISO is usually applied in 1 µM, or sometimes even in 10 µM concentrations. Under physiological conditions during a moderate sympathetic nervous system activation (e.g. during a physical exercise in medium heart rate reserve zones), the beta receptors are definitely not saturated.
According to various studies, the EC50 value of ISO on increasing ionic currents or sarcomere length shortening is in the range of 5-30 nM in a wide range of animal and cellular models [doi: 10.1152/ajpheart.433.2008, doi: 10.1038/sj.bjp.0701268, doi: 10.1016/0022-2828(89)90803-1, doi: 10.1016/S0022-2828(88)80121-4]. We were aiming to achieve a biologically relevant moderate beta-receptor activation, without saturating them.
To point out these in the text, the following sentence was inserted into the first paragraph of the Results section (line97-98): “10 nM ISO was used to achieve a biologically relevant moderate beta-receptor activation, without saturating them.”
Also, the cited sentence was rephrased to (line 183-184): “In the present study the effect of 10 nM ISO, corresponding to a moderate sympathetic activation of the heart […]”
- Was the effect of ISO on PKA activity measured? Has forskolin any effect on IK1 in this model?
We did not measure PKA and/or CaMKII activity directly. We did not use forskolin in our model, therefore we cannot explicitly state how would forskolin affect IK1 in this model.
- The significant limitation of the present study is that the role of PKA and CaCMK was assessed using only single inhibitors of these kinases. Are these inhibitors, at the doses applied, sufficiently specific to discriminate between them?
The initial report on H-89 [PMID: 2156866] determined its dissociation constant for PKA to be 0.048 nM in a cell-free in vitro assay, and H-89 is usually used at concentrations of 1-10 μM for PKA blockade [doi: 10.1016/j.yjmcc.2007.05.022., doi: 10.1152/ajpcell.1995.268.3.C651, doi: 10.1007/s00395-021-00850-2]. We avoided using higher H-89 concentrations to prevent potential non-specific CaMKII blockade, which could occur due to H-89 interacting with the ATP binding site of CaMKII. Indeed, H-89 has been shown to inhibit CaMKII with a dissociation constant of approximately 30 μM in cell-free in vitro assays [PMID: 2156866]. However, in pheochromocytoma cell line lysates, 30 μM H-89 did not significantly reduce CaMKII activity [PMID: 2156866]. It is important to note that the effect of H-89 heavily depends on substrate and ATP concentrations [doi: 10.1042/0264-6021:3510095], as well as the presence of potential activators [doi: 10.1152/ajpcell.1995.268.3.C651]. Additionally, pheochromocytoma cells primarily express the neuronal CaMKIIα isoform, while CaMKIIδ is the major cytosolic isoform found in cardiac cells, which may exhibit different affinity for H-89.
Besides this, H-89 is shown to be a potent and selective inhibitor of β1 and β2 adrenergic receptor ligand binding with respective Ki values of around 350 nM and 180 nM on human airway smooth muscle cells [PMID: 9918542].
With regards to KN-93, we did not find any reports on potential PKA inhibitory effect. However, when applied extracellularly, 1 µM KN-93 has been shown to abolish IKr in rabbit and guinea pig ventricular myocytes [doi: 10.1016/j.yjmcc.2015.10.012].
Based on the above-mentioned considerations, the kinase inhibitors were applied in relatively low concentrations (1 µM KN-93 and 3 µM H-89), and in the pipette solution.
The Methods section of the article was supplemented with a brief summary of these information (lines 391-395):
“When applied extracellularly, 1 µM KN-93 has been shown to abolish IKr in rabbit and guinea pig ventricular myocytes [46], whereas H-89 is shown to be a potent and selec-tive inhibitor of β1 and β2 adrenergic receptor ligand binding with respective Ki values of around 350 nM and 180 nM on human airway smooth muscle cells [39]. Therefore, both kinase inhibitors (1 µM KN-93 and 3 µM H-89) were applied in the pipette solution.”
Round 2
Reviewer 2 Report
Comments and Suggestions for Authors
The manuscript has been revised according to the reviewers' comments.